# The Initial Performance Evaluation of Mixed Multi-Frequency Undifferenced and Uncombined BDS-2/3 Precise Point Positioning under Urban Environmental Conditions

**Fuxin Yang** *, **Chuanlei Zheng, Jie Zhang, Zhiguo Sun, Liang Li and Lin Zhao**

College of Intelligent Systems Science and Engineering, Harbin Engineering University, Harbin 150001, China
* Correspondence: yangfuxin@hrbeu.edu.cn

**Abstract:** With the full operation of the global BeiDou navigation satellite system (BDS-3), positioning performance can be further enhanced by BDS-3 combined with the regional BeiDou navigation satellite system (BDS-2). However, due to satellite signals being out of lock and the limited visibility of satellites, the traditional multi-frequency BDS-2/3 precise point positioning (PPP) model is unable to maintain great positioning performance under urban environmental conditions. In this study, a mixed multi-frequency undifferenced and uncombined (UDUC) BDS-2/3 PPP model is presented to improve the positioning performance under urban environmental conditions by making full use of B1I, B1C, B2I, B2a, and B3I signals from all visible BDS satellites. In this model, BDS satellites with single-, dual-, triple- and quad-frequency observations all can participate in PPP. The static and kinematic experiments were carried out using the mixed multi-frequency UDUC BDS-2/3 PPP model to fully assess the positioning performance under urban environmental conditions with comparisons to the multi-frequency model. The static experiments indicated that the mixed multi-frequency UDUC BDS-2/3 PPP could continuously achieve decimeter-level positioning accuracy at a cut-off elevation angle of 40°, but part of the BDS-3 PPP would lose resolution due to limited visible satellites. Furthermore, the initial kinematic vehicle experiment showed that mixed multi-frequency UDUC BDS-2/3 PPP had better satellite geometry and more observation redundancy than the traditional multi-frequency model. Compared with the traditional multi-frequency BDS-2/3 model, the positioning accuracy of the mixed multi-frequency model was improved by 51.6, 35.5, and 39.1%, respectively, in east, north, and up directions. The convergence time was shortened by 40%.

**Keywords:** mixed multi-frequency; precise point positioning; BDS-2/3; urban environment



## 1. Introduction

Precise point positioning (PPP) is one of the precise positioning techniques based on the Global Navigation Satellite System (GNSS); the positioning accuracy can be achieved from decimeter-level to millimeter-level at a single user receiver worldwide without ground reference station assistance when the precise satellite orbits and clocks are used [1] for positioning [2,3]. To satisfy the demand for rapid social development, global positioning system (GPS) PPP-based solutions can be critical to applications in maritime, air, railway, and even automotive driving for highly precise positioning in areas without ground reference stations, such as RTX of Trimble, StarFix of Fugro and StarFire of NavCom [4–6]. However, the positioning accuracy of PPP will be degraded or even interrupted under urban environmental conditions where satellite occlusion is more severe [7].

As reported in the previous studies, due to the limited visibility of satellites under urban environmental conditions, the positioning accuracy of the dual-frequency ionosphere-free (IF) regional BeiDou navigation satellite system (BDS-2) PPP under urban environmental conditions is 0.53, 0.32 and 0.67 m in the east, north and up components, respectively [8]. Meanwhile, Guo, et al. found that B1I and B2I are in the presence of poor tracking and

contamination under urban environmental conditions, which decreased the positioning accuracy of multi-frequency combined BDS PPP [9]. With the full operation of global BeiDou navigation satellite system (BDS-3), Zhao, et al. designed the shielding experiments to evaluate the performance of undifferentiated and uncombined (UDUC) BDS-3 PPP under urban environmental conditions [10]. However, only B1c, B2b, and B2a signals were adopted, and the decimeter-level positioning of BDS-3 was supported at the highest cut-off elevation angle of 30 degree. Lu, et al. found that the B2a signal had the best anti-multipath performance, and the B1C signal had the worst capability [11]. Therefore, the urban environment is a challenge to BDS-2 and BDS-3 PPP due to the limitations of visible satellites, and loss of lock may occur for BDS PPP due to receiver dynamics, multipath, and interference for a given receiver configuration in a complex environment [12].

It Is widely accepted that the multi-constellation and multi-frequency combination is effective in enhancing positioning performance due to the better observation redundancy and satellite geometry [13–15]. Different from other systems of GNSS, not only are there 30 BDS-3 satellites that have been launched for global positioning, navigation, and timing (PNT) services, but there are also the 15 BDS-2 satellites that will continue to serve the whole Asian Pacific region for the next few years [16]. Besides the existing B1I, B2I, and B3I signals of BDS-2, the new signals B1C/B2a of BDS-3 have been designed to broadcast [17]. The satellite signal and orbit types for BDS are shown in Table 1 [18]. Since BDS itself has the advantages of multi-frequency and multi-constellation, BDS-2 joint BDS-3 can provide sufficient satellites including geostationary earth orbit (GEO), medium earth orbit (MEO), and inclined geosynchronous orbit (IGSO) satellites, and frequency points for a highly precise positioning service under urban environmental conditions.

**Table 1.** Satellite signal and orbit types for BDS.

| System | Constellation | PRN | Signals |
|--------|---------------|-----|---------|
| BDS-2 | GEO | C01-C05 | B1I, B2I, B3I |
| | IGSO | C06-C10/C13/C16 | B1I, B2I, B3I |
| | MEO | C11/C12/C14 | B1I, B2I, B3I |
| BDS-3 | GEO | C59-C61 | B1I, B1C, B2a, B2b, B3I |
| | IGSO | C38-C40 | B1I, B1C, B2a, B2b, B3I |
| | MEO | C19-C30/C32-C37/C41-C46 | B1I, BI1C, B2a, B2b, B3I |

When dual-frequency BDS-3 satellites are incorporated with BDS-2, the convergence time reduces and the positioning accuracy improves [19]. With the emergence of real-time precise satellite products, the dual-frequency ionosphere-free BDS-2/3 PPP has been analyzed and the centimeter-level accuracy can be obtained in ambiguity float mode after convergence based on CLK93 real-time stream [20]. However, the relatively poor availability of specific dual-frequency observations can cause a worsening of the satellite geometry and the reduction of observation redundancy, even the loss of the PPP positioning solution [21]. Thus, in order to improve observation redundancy, some scholars presented a multi-frequency BDS-2/3 PPP model to further improve the accuracy and convergence time [22–26]. Compared with dual-frequency PPP, the accuracy of the five-frequency UDUC BDS-2/3 PPP model improves by 21.4, 0, and 5.6%, respectively, in East, North, and Up directions, and the convergence time is shortened by 16.2% [27]. The existing research mainly focused on the multi-frequency BDS-2/3 PPP model based on the good quality data of static stations, and the impact of satellite signals out of lock and limited visible satellites under urban environmental conditions were not considered. For the traditional multi-frequency model, once the satellites lack a specific frequency signal, this satellite will not participate in PPP. In order to solve the problem and make full use of all available observations, a mixed frequency model was presented based on the characteristics of the undifferenced and uncombined (UDUC) model [21,28]. Li, et al. proposed a mixed single- and dual-frequency quad-constellation precise point positioning model. In the kinematic test, the accuracy improvement rates reached 78 and 76% over the traditional

dual-frequency PPP, and 13 and 38% over the single-frequency PPP, respectively [29]. The mixed frequency model can effectively improve the positioning performance under urban environmental conditions, but the current research of mixed frequency models mainly focuses on data with single- and dual-frequency observations [30]. Therefore, the positioning performance of the mixed multi-frequency BDS-2/3 under urban environmental conditions where satellite and signal occlusion are more severe should be further verified.

In this contribution, we present the mathematical models of the mixed multi-frequency UDUC BDS-2/3 PPP. Different from the traditional multi-frequency model, the BDS satellites with single-, dual-, triple- and quad-frequency observations all can participate in PPP in the mixed multi-frequency model. Thus, the satellite geometry and observation redundancy of the mixed multi-frequency model are always better under urban environmental conditions. Secondly, after a brief statement about the data processing strategy, the performances of the mixed multi-frequency UDUC BDS-2/3 PPP are initially analyzed by the static data at different cut-off elevation angles and by kinematic data under the real urban environmental conditions, respectively. Meanwhile, the comparative experiments of the mixed multi-frequency model and the traditional multi-frequency model were designed to verify the advantages of the mixed multi-frequency model under urban environmental conditions. Finally, the discussion and conclusions are drawn.

## 2. Methodology

The relationship of signal frequency for BDS is shown in Figure 1, in which B2b is a satellite-based enhancement signal, so BDS including BDS-2 and BDS-3 consists in total of five frequency signals, namely B1I, B1C, B2I, B2a, and B3I. The observations of the UDUC PPP model can be simplified as [31]:

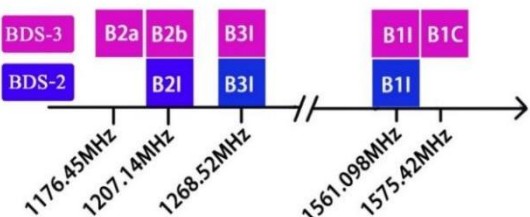

**Figure 1.** BDS signals in the L band.

$$L_{i,g}^{j,s} = \rho_i^{j,s} + dt_i^s - dt^{j,s} - \gamma_{i,g} \cdot I_1^{j,s} + T_i^{j,s} + \lambda_g(N_{i,g}^{j,s} + b_{i,g}^{j,s}) + \xi_{i,g}^j \tag{1}$$

$$P_{i,g}^{j,s} = \rho_i^{j,s} + dt_i^s - dt^{j,s} + \gamma_{i,g} \cdot I_1^{j,s} + T_i^{j,s} + d_{i,g}^{j,s} + \varepsilon_{i,g}^{j,s} \tag{2}$$

where $L$ and $P$ respectively express carrier phase and pseudo-range observations; The superscript $s$ representing one of BDS-2 or BDS-3, $g$ ($g$ = B1I, B1C, B2I, B2a, B3I) is the frequency of BDS satellites; $\rho_i^j$ is the geometric distance between the phase center of satellite $j$ and rover $i$, $dt_i$ and $dt^j$ indicate the receiver and satellite clock bias, respectively; $I_{i,1}^j$ is the slant ionosphere refraction delay on $f_1$, $\gamma_g = f_1^2/f_g^2$; $T_i^j$ is wet tropospheric refraction delay; $\lambda_g$ denotes the wavelength at $f_g$; $d_{i,g}^j = d_{i,g} - d_g^j$ is the difference of code biases between receiver and satellite at $f_g$; $\varepsilon_{i,g}^j$ and $\xi_{i,g}^j$ are, respectively, unmodeled errors for the pseudo-range and carrier phase observations, mainly including multipath effect and observation noise. The units of the above parameters are meters. $N_{i,g}^j$ is the integer ambiguity at $f_g$; $b_{i,g}^j = b_{i,g} - b_g^j$ is the difference of uncalibrated phase delays (UPD) between receiver and satellite at $f_g$. The unit of integer ambiguity and UPD is in cycles. In addition, the errors which are not mentioned in Equations (1) and (2), including phase wind-up delay, relativity effects, site displacement effects and tide effects, are corrected by empirical models [32].

The biases of the mixed multi-frequency UDUC BDS-2/3 model need to be properly handled to improve positioning accuracy and shorten convergence time. Therefore, the

expressions about DCB between code biases in different frequencies, the inter-frequency bias (IFB) between different DCBs, and the inter-system bias (ISB) in combined BDS-2/3 are derived. For convenience, the following notations are defined:

$$\begin{cases} \alpha_{mn} = \frac{f_m^2}{f_m^2 - f_n^2}, \; \beta_{mn} = -\frac{f_n^2}{f_m^2 - f_n^2} \\ DCB_{mn}^j = d_m^j - d_n^j, \; DCB_{i,mn} = d_{i,m} - d_{i,n} \end{cases} \tag{3}$$

where $DCB_{mn}^j$ and $DCB_{i,mn}$ are differential code bias (DCB) between frequency $m$ and $n$ for satellite and receiver, respectively. The unit of DCB is in meters.

Normally, the precise satellite orbit and clock products of the International GNSS Service (IGS) Multi-GNSS Experiment (MGEX) are utilized to correct satellite orbit and clock errors [33]. Meanwhile, B1I/B3I ionosphere-free combination is used to obtain the precise BDS satellite clock corrections. For this reason, the correction parameter $dt_{13}^j$ of the satellite clock can be depicted as:

$$dt_{13}^j = dt^j + (\alpha_{13} \cdot d_{i,1}^j + \beta_{13} \cdot d_{i,3}^j) \tag{4}$$

In the mixed multi-frequency UDUC PPP model, the effects of DCBs on the B1C, B2I, B2a pseudo-range cannot be completely merged with the ionospheric delay. Therefore, the additional IFB parameter associated to (4) needed to be estimated to make up the effects, and the IFBs of B1C, B2a and B2I can be written as:

$$ifb_g = \frac{\beta_{13}}{\beta_{1g}} DCB_{i,13}^j - DCB_{i,1g}^j \tag{5}$$

Although the BDS-2 and BDS-3 belong to BDS, ISB still exists in BDS-2/3 PPP. When IGS precision clock difference products are used, ISB includes not only the receiver code biases between BDS-2 and BDS-3 but also the reference satellite clock difference. Therefore, it can be depicted as:

$$isb = \alpha_{13}(d_{i,1}^{BDS-3} - d_{i,1}^{BDS-2}) + \beta_{13}(d_{i,3}^{BDS-3} - d_{i,3}^{BDS-2}) + (D_{IGS}^{BDS-3} - D_{IGS}^{BDS-2}) \tag{6}$$

where $D_{IGS}$ is the precision satellite clock datum from IGS in this paper.

Assuming that the receiver has received $n$ carrier phase observations, they are paired with pseudo-range observations. Therefore, the linearized measurement equation of the mixed multi-frequency UDUC BDS-2/3 PPP can be expressed as:

$$\begin{bmatrix} P_{uduc} \\ L_{uduc} \end{bmatrix} = \begin{bmatrix} G & M_{RC} & M_T & M_I & M_{IFB} & 0 \\ G & M_{RC} & M_T & -M_I & 0 & I \end{bmatrix} \cdot \begin{bmatrix} X \\ clk \\ d_w \\ I_{BIC} \\ IFB \\ N \end{bmatrix} + \begin{bmatrix} \varepsilon_{uduc} \\ \xi_{uduc} \end{bmatrix} \tag{7}$$

$$R = \left( k_s \cdot \begin{bmatrix} \left(\sigma_{P_g^j}^j\right)^2 & 0 \\ 0 & \left(\sigma_{L_g^j}^j\right)^2 \end{bmatrix} \right) \otimes I \tag{8}$$

where $P_{uduc}$ and $L_{uduc}$ are the $n \times 1$ UDUC observations, respectively. $X = [x, y, z]^T$ are the estimated coordinates with $n \times 3$ design matrix $G$. $clk = [clk^{BDS3}, isb^{BDS3-BDS2}]^T$ includes receiver clock errors and ISB parameters with $n \times 2$ matrix $M_{RC}$. The first column of $M_{RC}$ is 1, and the second column is 1 when the corresponding observations belong to BDS-2. Otherwise, it is 0. $d_w$ is the zenith wet delay of the troposphere with $M_T$ derived from the Global Mapping Function model [34]. $M_I$ is the $n \times 1$ design matrix related to $I_{B1I}$ from Equation (1). $IFB = [ifb_{B1C}, ifb_{B2I}, ifb_{B2a}]^T$ are the inter-frequency bias related to the ionosphere-free combination of B1I/B3I with $n \times 3$ matrix $M_{IFB}$. The element in the first column of $M_{IFB}$ is 1 for B1C, the element is 1 in the second column for B2I, the element is 1

in the third column for B2a; the remaining elements are 0. $N$ is the $n \times 1$ real ambiguity vector and $I$ is the identity matrix with rank $n$. $\varepsilon_{uduc}$ and $\zeta_{uduc}$ are the $n \times 1$ unmodeled errors for the pseudo-range and carrier phase observations, respectively. $\sigma_P$ and $\sigma_L$ are the STDs of UDUC pseudo-range and carrier phase noise, respectively. The unmodeled errors of pseudo-range and carrier phase multipath are represented by the elevation angle model [35]. $R$ is the variance–covariance for the unmodeled errors and can be expressed as Equation (8). $\otimes$ is the Kronecker product operator. With the derived mathematical and stochastic models based on Equations (7) and (8), the position coordinates can be calculated based on the robust sequential least squares model [13]. Above all, Table 2 provides the detailed strategy of the mixed multi-frequency UDUC BDS-2/3 PPP.

**Table 2.** Mixed multi-frequency UDUC BDS-2/3 PPP strategy.

| Items | Strategies |
|---|---|
| Signal frequency | BDS-3: B1I/B1C/B2a/B3I; <br> BDS-2/3: B1I/B1C/B2I/B2a/B3I; |
| Elevation cutoff | $10°, 20°, 30°, 40°$; |
| Observation weighting | The a priori precision of pseudo-range and carrier phase is 0.6m and 0.006m, respectively. As a result of the comparatively lower accuracy of the orbit and clock data of BDS GEO satellites, their weight values are reduced by 100 times; |
| Receiver coordinate | Estimated as white noise process; |
| Receiver clock offset | Estimated as white noise process; |
| Ionospheric delay | Estimated as white noise process; |
| Tropospheric wet delay | Estimated as random walk process; |
| Inter-frequency bias | Estimated as random walk process; |
| Inter system bias | Estimated as constant; |
| Phase ambiguities | Estimated as constant; float values; |

## 3. Experiment and Discussion

In order to evaluate the performance advantages of the mixed multi-frequency UDUC BDS-2/3 PPP under urban environmental conditions, the indicators of positioning accuracy and convergence time were evaluated in both static and kinematic experiments. In the static experiment, the positioning performances of the mixed multi-frequency UDUC BDS-2/3 PPP at different cut-off elevation angles were analyzed to show the advantages of combined BDS-2/3 under limited visible satellites. In the kinematic experiment, the advantages of the mixed multi-frequency model were verified by comparing with the traditional multi-frequency model. The positioning errors were defined as the differences between PPP solutions and reference solutions. The positioning accuracy was calculated based on the RMS of the positioning errors after solutions converge. The convergence time is defined as the time that the positioning errors are less than 0.1 m for continuous 20 epochs. The MG-APP was used for secondary development to complete the performance analysis of the mixed multi-frequency UDUC BDS-2/3 PPP model [36]. The processing strategy of the mixed multi-frequency UDUC BDS-2/3 PPP is shown in Table 2.

### 3.1. Static Experiment

The datasets with 30 s sampling interval were collected from JFNG, NNOR, MIZU, SGOC, and IISC station to assess the performances of the mixed multi-frequency UDUC BDS-3 and BDS-2/3 PPP. The locations of selected stations are presented in Figure 2 and the data can be obtained from the website (ftp://igs.bkg.bund.de/IGS/ (access on 9 March 2022)). Meanwhile, in the static experiment, the reference coordinates of the stations were obtained from the website (ftp://igs.ign.fr/pub/igs/products (access on

9 March 2022)). The information from stations for the static experiment is shown in Table 3; the reason for selecting these stations was that five frequency signals of BDS-2/3 could be received. Meanwhile, the precise satellite ephemeris and clock data could be obtained from the website (ftp://ftp.gfz-potsdam.de/pub/GNSS/products/mgex (access on 9 March 2022)).

**Table 3.** Station information for the static experiment.

| No | Station | Location | Receiver | Frequency of BDS-2/3 |
|----|---------|----------|----------|----------------------|
| 1 | JFNG | China | TRIMBLE ALLOY | B1I, B2I, B3I, B1C, B2a |
| 2 | NNOR | Australia | SEPT POLARX5TR | B1I, B2I, B3I, B1C, B2a |
| 3 | MIZU | Japan | SEPT ASTERX4 | B1I, B2I, B3I, B1C, B2a |
| 4 | SGOC | Sri Lanka | JAVAD TRE_3 | B1I, B2I, B3I, B1C, B2a |
| 5 | IISC | India | SEPT POLARX5 | B1I, B2I, B3I, B1C, B2a |

There is almost no signal frequency loss due to the good data quality of the static station; thus, the model of the mixed multi-frequency UDUC PPP is similar to the traditional multi-frequency model. Therefore, the performance comparison between the traditional multi-frequency model and the mixed multi-frequency model is omitted in this section. The static experiment set the cut-off elevation angles of the mixed multi-frequency model in steps of 10° to simulate satellite occlusion conditions under urban environmental conditions. When the cut-off elevation angle reaches 50°, the sampling data of both BDS-3 and BDS-2/3 cannot meet the number of satellites required by PPP. Thus, the positioning performance was analyzed with the cut-off elevation angle from 10 to 40°. In static experiments, the positioning performances of the mixed multi-frequency UDUC BDS-3 and BDS-2/3 were analyzed and compared. Firstly, the 24 h data of JFNG station on day of year (DOY) 035 were taken to evaluate positioning accuracy and convergence time between BDS-3 and BDS-2/3 at different cut-off elevation angles, as shown in Figure 3.

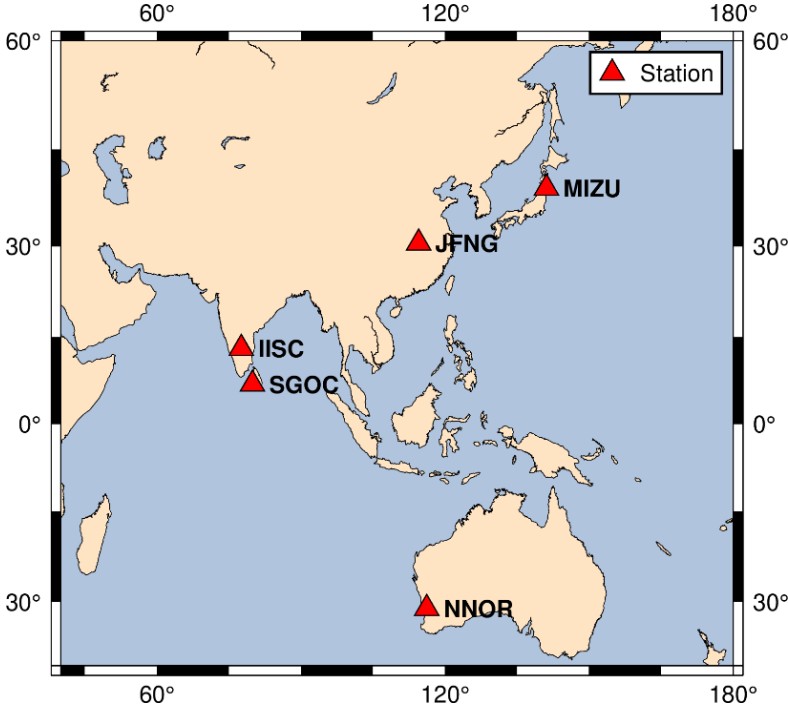

**Figure 2.** The location of selected stations for static experiments.

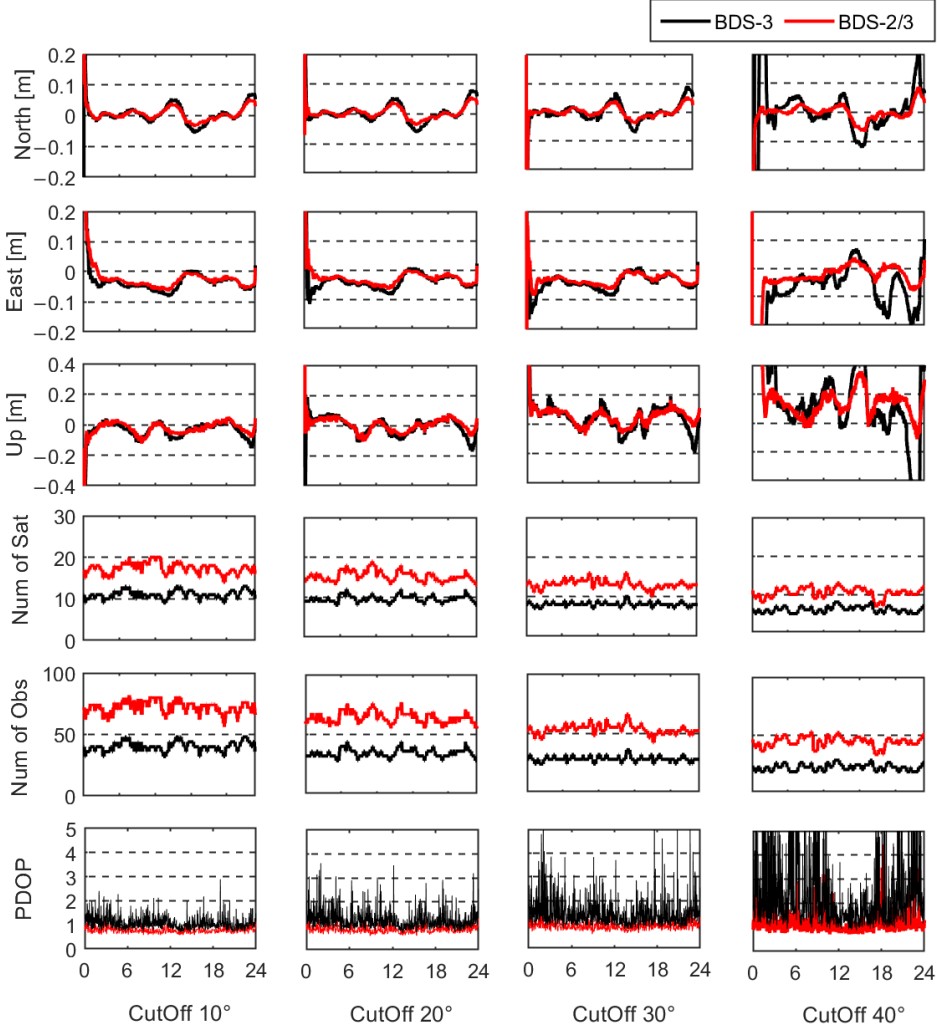

**Figure 3.** Positioning errors, number of satellites, number of carrier-phase observations and PDOP of JFNG station at different cut-off elevation angles.

Figure 3 shows the positioning error series, the number of satellites, the total number of carrier-phase observations, and PDOP at different cut-off elevation angles. It can be clearly observed that the positioning errors of BDS-2/3 were less than BDS-3 at different cut-off elevation angles, and the positioning errors of BDS-3 and BDS-2/3 increased with the increasing cut-off elevation angle. Meanwhile, when the cut-off elevation angle reached 40°, the positioning stability of BDS-3 significantly decreased. The maximum horizontal positioning error of BDS-3 was close to 0.2 m and the maximum vertical positioning error exceeded 0.5 m. Under this condition, the positioning error series of BDS-2/3 was obviously more stable and the maximum horizontal and vertical positioning errors were less than 0.1 m and 0.4 m, respectively. The cause of the above phenomena is that BDS-2/3 has more available satellites and better geometric distribution than BDS-3. Although the cut-off elevation angle reaches 40°, there are still an average of 10.47 available satellites and the PDOP is 1.13 for BDS-2/3. The number of satellites and observations for BDS-2/3 at the cut-off elevation angle of 40° is similar to BDS-3 at the cut-off elevation angle of 10°. The statistical results of positioning errors and convergence time for BDS-3 and BDS-2/3 are given in Table 4. Compared with BDS-3, the positioning errors of BDS-2/3 at all cut-off elevation angles were reduced by more than 20.45 % in the horizontal direction and more than 9.57 % in the vertical direction, respectively.

**Table 4.** Positioning errors (m) and convergence time (h) at JFNG station at different cut-off elevation angles.

| Elevation Angle | Direction | BDS-3 | BDS-2/3 | Improvement |
|---|---|---|---|---|
| 10° | East | 0.044 m | 0.035 m | 20.45% |
| | North | 0.027 m | 0.019 m | 29.63% |
| | Up | 0.052 m | 0.040 m | 23.08% |
| | CT | 0.38 h | 0.33 h | 13.16% |
| 20° | East | 0.047 m | 0.036 m | 23.40% |
| | North | 0.029 m | 0.019 m | 34.48% |
| | Up | 0.056 m | 0.040 m | 28.57% |
| | CT | 0.27 h | 0.17 h | 37.04% |
| 30° | East | 0.051 m | 0.037 m | 27.45% |
| | North | 0.030 m | 0.020 m | 33.33% |
| | Up | 0.115 m | 0.104 m | 9.57% |
| | CT | 0.41 h | 0.36 h | 12.20% |
| 40° | East | 0.082 m | 0.036 m | 56.10% |
| | North | 0.055 m | 0.026 m | 52.73% |
| | Up | 0.261 m | 0.161 m | 38.31% |
| | CT | 3.36 h | 1.52 h | 54.76% |

Likewise, as is shown in Table 4 that the convergence time of BDS-2/3 was shorter than BDS-3 at all cut-off elevation angles. Note that CT in Table 4 is the abbreviation for convergence time. It is worth noting that the convergence time at 20° was shorter than 10° for BDS-3 and BDS-2/3. The reason is that the value of PDOP is similar at cut-off elevation angles of 10 and 20°, and the cut-off elevation angle of 20° can additionally shield satellites of poor quality with an elevation angle of 10 to 20°. In addition, with the cut-off elevation angle increasing, the number of satellites reduces and the convergence time increases significantly. The longest convergence time of BDS-2/3 was 1.61 h, which took about five times as long as the time when the cut-off elevation angle is 10°. Compared with BDS-3, the convergence time of BDS-2/3 at all elevation angles was shortened by more than 10%. The improvement was largest when the cut-off elevation angle was 40 °, and its value was 54.76%.

To avoid the contingency of results based on a single station and a single day, the observation data of the JFNG, NNOR, MIZU, SGOC, and IISC stations from DOY 032 to 038 in 2021 were analyzed. Figure 4 depicts the positioning errors at different cut-off elevation angles for BDS-3 and BDS-2/3. In the boxplots, each box has five horizontal lines, which are maximum, upper quartile, median, lower quartile, and minimum from top to bottom. It should be mentioned that when the cut-off elevation angle reached 40°, due to the limitation of the number of satellites, BDS-3 could not maintain continuous positioning in more than half of the sampled observation data. Thus, the boxplots show the positioning errors of the data which met the PPP positioning conditions, and the positioning accuracy of BDS-3 PPP was less than BDS-2/3 at the cut-off elevation angle of 40°. At all cut-off elevation angles, the minimum errors of BDS-2/3 were similar to BDS-3. It can be concluded that the positioning accuracy of the two combines after convergence is approximate. However, the maximum errors of BDS-2/3 were all much less than BDS-3. Compared with BDS-3, the improvement of BDS-2/3 at 40° is the most significant, the maximum errors are reduced by 0.058, 0.077, and 0.102 m, respectively, in E, N, and U directions.

Table 5 shows the medians of positioning errors at different cut-off elevation angles. It can be seen that with the increasing cut-off elevation angles, the horizontal errors of BDS-2/3 and BDS-3 increased slightly. However, their vertical errors increased significantly. At the cut-off elevation angle of 40°, the medians of the horizontal errors for BDS-2/3 were within 0.035 m, and the vertical errors increased to 0.218 m. What is more, compared with BDS-3, the medians of the positioning errors for BDS-2/3 were all reduced, by 24.00 to 43.59% in the horizontal direction and 12.45 to 33.33% in the vertical direction, respectively. To sum up, the positioning accuracy of the mixed multi-frequency BDS-2/3 PPP was still better than BDS-3 in the case of a large amounts of data.

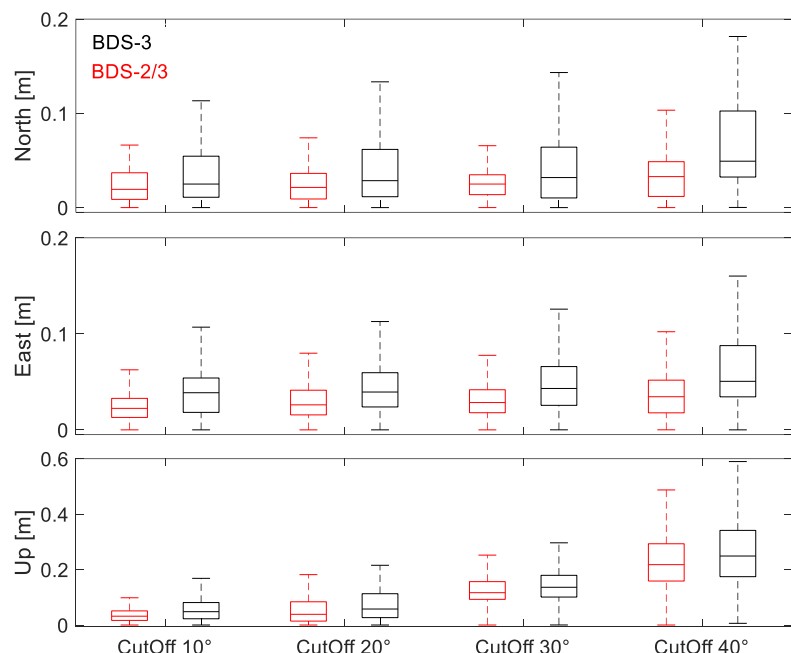

**Figure 4.** The positioning errors in E, N, U directions at JFNG, NNOR, MIZU, SGOC, and IISC stations.

**Table 5.** Medians of the positioning errors (m) at different cut-off elevation angles.

| Elevation Angle | Direction | BDS-3 | BDS-2/3 | Improvement |
|---|---|---|---|---|
| 10° | East | 0.039 m | 0.022 m | 43.59% |
| | North | 0.025 m | 0.019 m | 24.00% |
| | Up | 0.048 m | 0.032 m | 33.33% |
| 20° | East | 0.039 m | 0.026 m | 33.33% |
| | North | 0.029 m | 0.021 m | 27.59% |
| | Up | 0.057 m | 0.039 m | 31.58% |
| 30° | East | 0.043 m | 0.028 m | 34.88% |
| | North | 0.034 m | 0.025 m | 26.47% |
| | Up | 0.136 m | 0.117 m | 13.97% |
| 40° | East | 0.050 m | 0.035 m | 30.00% |
| | North | 0.049 m | 0.033 m | 32.65% |
| | Up | 0.249 m | 0.218 m | 12.45% |

In order to assess convergence time, we simulated reconvergence every 6 h by artificially initializing parameters. Figure 5 shows the cumulative distribution function of BDS-2/3 and BDS-3 at different cut-off elevation angles. As we can see, the convergence times at different cut-off elevation angles of BDS-2/3 were all less than BDS-3. Meanwhile, with the increasing cut-off elevation angle, the improvement of convergence time by BDS-2/3 was gradually enhanced. Compared with BDS-3, the maximum convergence times of BDS-2/3 shortened by 32.5, 43.1, and 52.3%, respectively, at 10, 20, and 30°. When the cut-off elevation angle was 40°, continuous PPP of BDS-3 could be achieved only in the minority of the sampled observation data. In addition, for a 68% confidence level, the convergence times of BDS-3 were 0.371 h, 0.720 h, 1.297h, and 2.201 h, respectively, with the increasing cut-off elevation angles, and the convergence times of BDS-2/3 were 0.242, 0.402, 0.697, and 1.392 h, respectively. Figure 6 shows the mean of convergence times for BDS-3 and BDS-2/3 at different cut-off elevation angles. At the cut-off elevation angle of 10, 20, 30, and 40°, the average convergence times of BDS-2/3 shortened by 40.06, 47.60, 47.82 and 38.86%. It can be concluded that BDS-2/3 can effectively shorten convergence time under urban environmental conditions with satellite occlusion when compared with BDS-3.

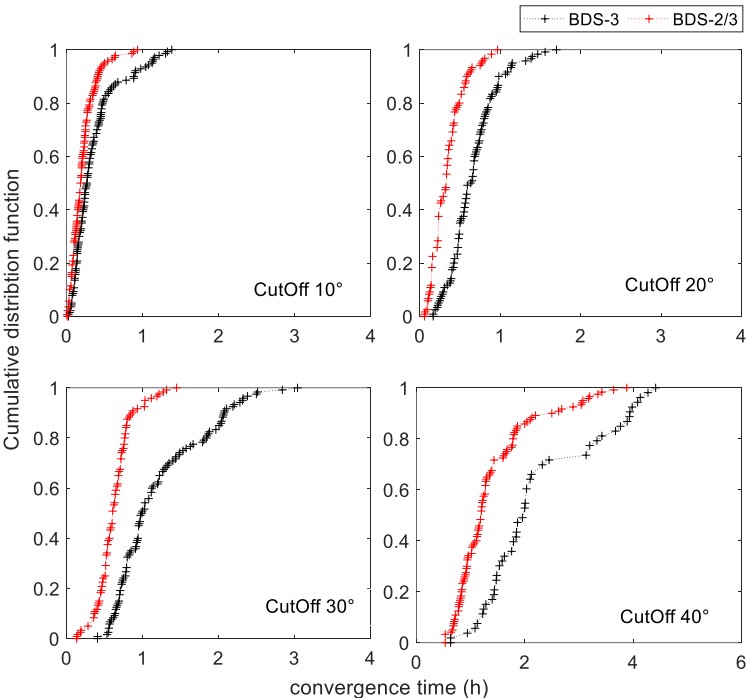

**Figure 5.** The convergence time comparison between the two combinations. The cumulative distribution function curve.

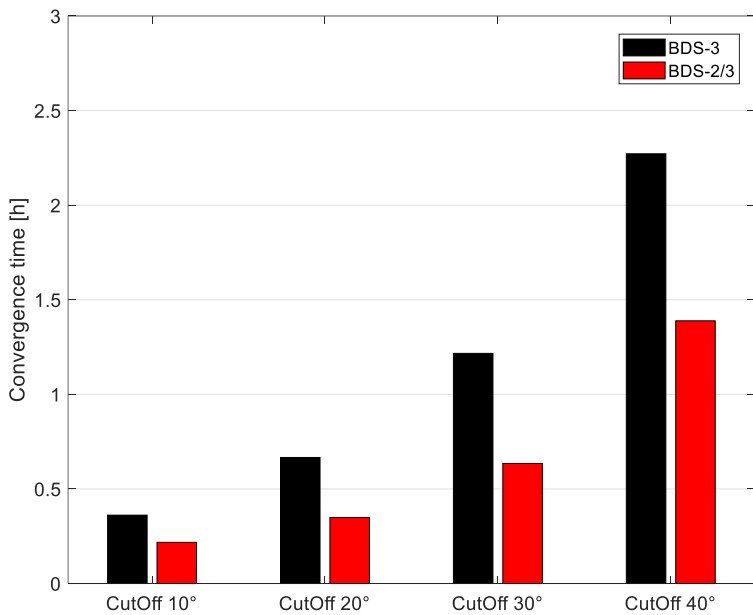

**Figure 6.** The mean of convergence time histograms.

### 3.2. Kinematic Experiment

To assess the mixed multi-frequency UDUC BDS-2/3 PPP performance in the kinematic experiment, real data with 1s interval was collected from Harbin Engineering University on DOY 258 in 2021. The data time span was nearly 3.5 h. Figure 7 shows the experimental device. A NovAtel "OEM729" GNSS receiver with a Harxon HX-CSX601A antenna was carried by a modified test vehicle as a rover station to collect kinematic data. The "OEM729" GNSS receiver can receive BDS B1I/B1c/B2a/B2I/B3I observations. An "OEM628E" GNSS receiver with a Novatel 750 antenna was installed at the top of the No. 16 apartment in Harbin Engineering University as the base station. For kinematic experiment,

the reference coordinates of the kinematic experimental device are usually obtained by RTKLIB, which is an open source software package for GNSS positioning and can be obtained from the website (https://www.rtklib.com/ (access on 9 December 2021)). In the paper, the observation data of the reference station and the kinematic experimental device was processed together by RTKPOST in RTKLIB and the centimeter-level fixed-solution obtained was used as a reference to evaluate the positioning performance. Figure 8 shows the real urban environmental conditions of the kinematic experiment; the experimental route covered the parking lot of Building No. 31, Harbin Engineering University and the trajectory of the kinematic experimental device is shown in Figure 8. The straight-line distance from the kinematic experimental device to the base station is about 1.1 km. The kinematic experimental device moved at a speed of around 20 km per hour and ran about 40 laps in the experimental site.

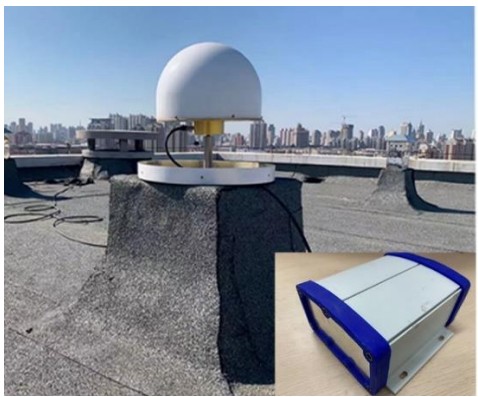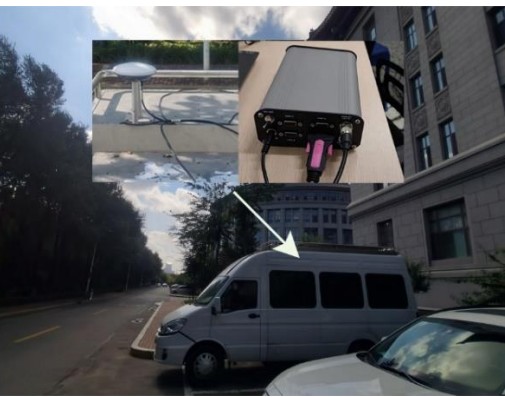

**Figure 7.** Schematic diagram of reference station (**left**) and vehicle-mounted kinematic experimental device (**right**).

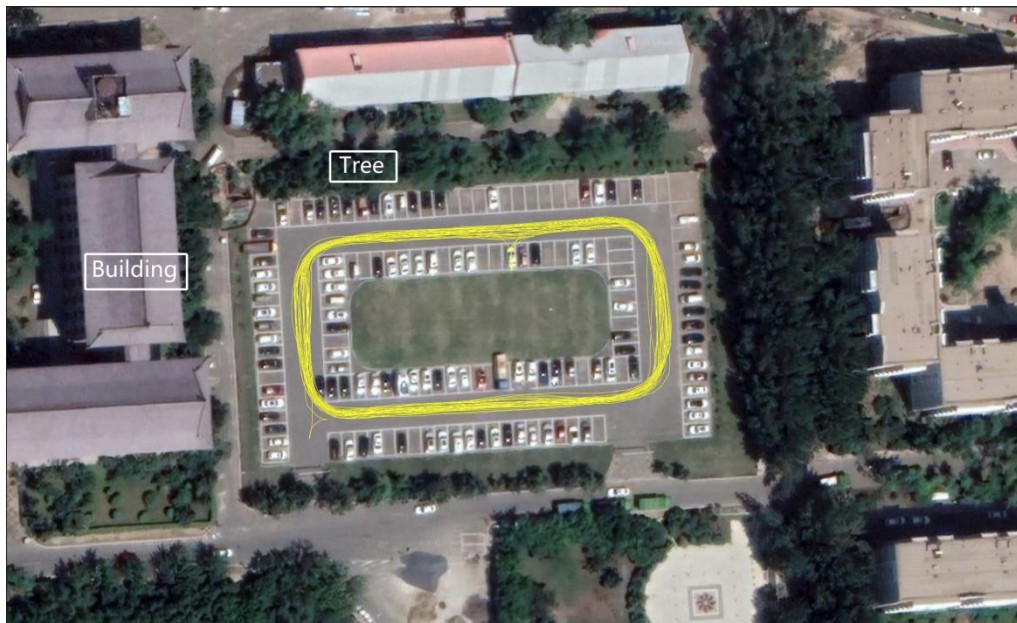

**Figure 8.** The motion trajectory of vehicle.

Compared with the data of the static experiment, the partial frequency signals of some satellites are lost due to the occlusion of surrounding buildings and trees, the nature satellite occlusion condition. In this case, the quality control strategy of a traditional multi-frequency model was to delete the satellites from which selected frequency signals were missing. However, for the mixed multi-frequency PPP model, as long as the satellite has one or more

frequency signals, the satellite will be retained and used for positioning. Therefore, the proposed model maximizes the use of visible satellites to reduce PDOP. In order to verify the advantages of the mixed multi-frequency model, the positioning performances of it were compared with the traditional multi-frequency model in the kinematic experiment.

Figure 9 shows the number of satellites, number of carrier-phase observations, and PDOP separately for kinematic PPP. Meanwhile, the mixed multi-frequency model possesses more carrier-phase observations and lower PDOP than the traditional multi-frequency model. Therefore, partial signals were lost during the kinematic experiment period, and the advantages of the mixed multi-frequency BDS-2/3 PPP model under urban environmental conditions could be verified. Compared with BDS-3, the number of satellites of BDS-2/3 is higher and the PDOP for BDS-2/3 is lower. Thus, the mixed multi-frequency BDS-2/3 model owns the best satellite geometry and the most observation redundancy among the four combinations.

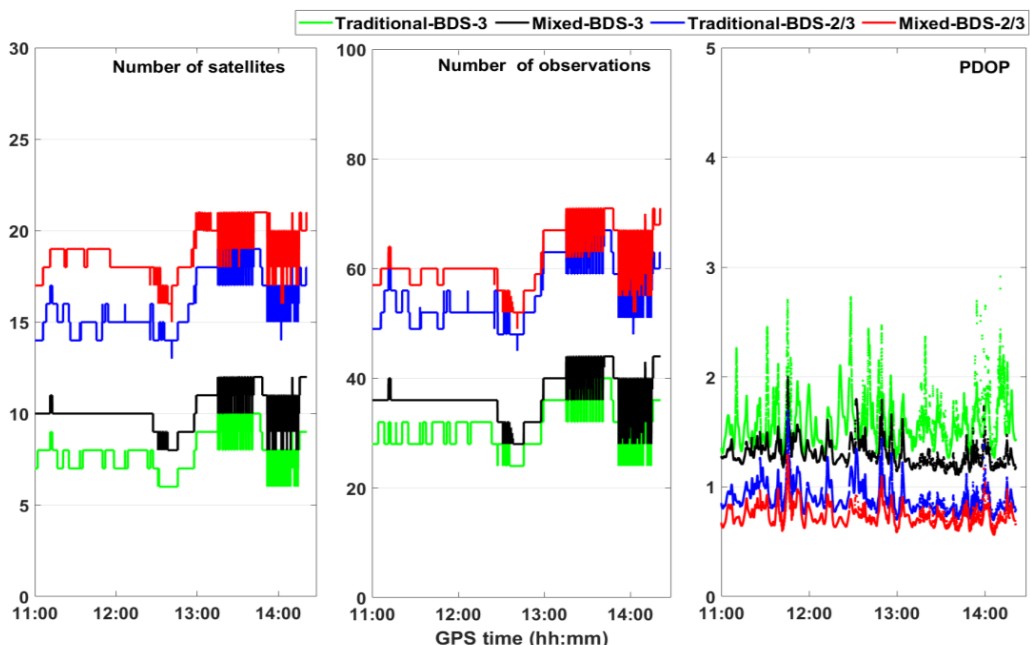

**Figure 9.** Number of satellites, number of carrier-phase observations and PDOP of kinematic PPP.

The positioning performance was compared between the traditional and mixed multi-frequency UCDC PPP, as shown in Figure 10. Compared with the traditional multi-frequency PPP, the positioning accuracy of the mixed multi-frequency PPP was better in kinematic conditions. The positioning errors in the E, N, and U directions of the mixed multi-frequency BDS-2/3 PPP were reduced by 38.6, 34.5, and 46.9%, respectively, and the positioning errors of the mixed multi-frequency BDS-3 PPP were reduced by 40.1, 49.5, and 39.2%, respectively. In addition, due to the best constellation geometry and observation redundancy among the four combinations, only mixed multi-frequency BDS-2/3 could maintain centimeter-level positioning accuracy in the E, N, and U directions. Meanwhile, as shown in Table 6, the convergence time of the mixed multi-frequency BDS-2/3 model was obviously the shortest, at 0.12 h. Compared with the convergence times of the traditional multi-frequency PPP, the convergence times of the mixed multi-frequency BDS-3 and BDS-2/3 PPP were shortened by 11.63 and 40.00%, respectively. To sum up, the mixed multi-frequency PPP model demonstrated a better positioning performance under urban environmental conditions.

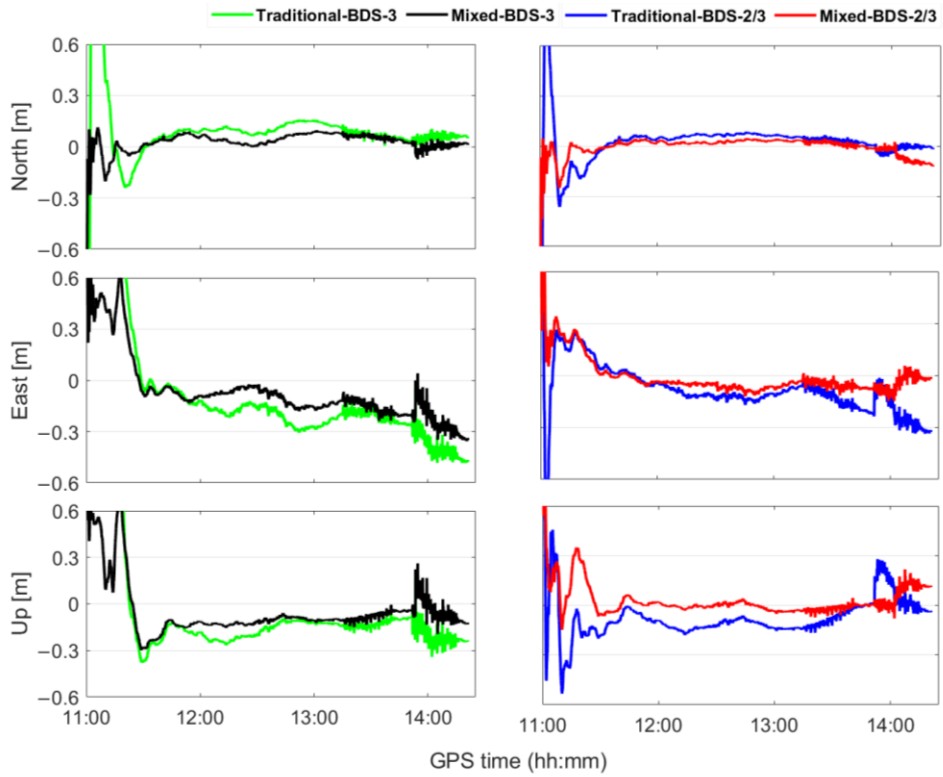

**Figure 10.** Mixed multi-frequency vs. traditional multi-frequency UDUC PPP positioning errors.

**Table 6.** The positioning errors in the E, N, and U directions and the convergence times of traditional multi-frequency and mixed multi-frequency UDUC PPP.

|       | Traditional BDS3 | Mixed BDS3 | Traditional BDS-2/3 | Mixed BDS-2/3 |
| ----- | ---------------- | ---------- | ------------------- | ------------- |
| North | 0.097 m          | 0.049 m    | 0.058 m             | 0.038 m       |
| East  | 0.287 m          | 0.172 m    | 0.132 m             | 0.081 m       |
| Up    | 0.245 m          | 0.149 m    | 0.128 m             | 0.068 m       |
| Time  | 0.43 h           | 0.38 h     | 0.20 h              | 0.12 h        |

## 4. Discussion

The BDS-3 PPP of the future will feature joint-use multi-frequency signals, and BDS-2 satellites will still operate in a few years. The BDS-2/3 PPP with multi-frequency combination is effective in enhancing positioning performance due to the better observation redundancy and satellite geometry. In the paper, we analyzed the positioning performance of the mixed multi-frequency BDS-2/3 PPP under the urban environmental conditions and validated the advantages of the mixed multi-frequency model. The 7 days of static data were set at different cut-off elevation angles to simulate satellite occlusion under urban environmental conditions. There was almost no signal frequency loss due to the good data quality of the static station; thus, the model of the mixed multi-frequency UDUC PPP is similar to the traditional multi-frequency model. The positioning accuracy and convergence time of BDS-2/3 were better than BDS-3 at different cut-off elevation angles. Compared with BDS-3, the medians of the positioning errors for BDS-2/3 were all reduced, by 24.00 to 43.59% in the horizontal direction and 12.45 to 33.33% in the vertical direction, respectively. When the cut-off elevation angle was 40°, the continuous PPP of BDS-3 could be achieved only in the minority of the sampled observation data. Furthermore, the real urban kinematic results showed that the mixed multi-frequency UDUC BDS-2/3 PPP could, respectively, achieve an accuracy of 0.081, 0.038, and 0.068 m in the E, N, and U directions, and the convergence time was 0.12 h. The partial signals were lost during the kinematic experiment period due to receiver dynamics, multipath, and interference for a given receiver configu-

ration. Compared with the convergence times of the traditional multi-frequency PPP, the convergence times of the mixed multi-frequency BDS-3 and BDS-2/3 PPP shortened by 11.63 and 40.00%, respectively. Compared with the traditional multi-frequency model, the mixed multi-frequency model could make full use of the observations of BDS, which is the main reason to improve positioning performance. Therefore, compared to the multi-frequency models mentioned in the papers of Li, et al. [25] and Wu, et al. [28], a mixed multi-frequency model which can make better use of BDS observations is more universal.

The kinematic performance was evaluated based on limited real data, and more kinematic experiments will be carried out in different urban environments in the future. Meanwhile, the mixed multi-frequency model in the paper is used for post-processing analysis, the mixed multi-frequency model capable of real-time data processing is the future development direction. In addition, the long convergence time is still a problem that cannot be ignored in practical applications. We expect to improve the convergence performance of BDS-2/3 PPP under urban environmental conditions through multi-GNSS and multi-frequency technology in future research, which can further improve the practicability of BDS-2/3 PPP in different fields.

## 5. Conclusions

Until recently, the PPP technique had been widely used to provide high-accuracy position solutions. However, urban environmental conditions with satellite and signal occlusion is a challenge for PPP application. The quality control strategy of a traditional multi-frequency model is to delete the satellites of which selected frequency signals are missing. In order to make full use of the multi-frequency B1I, B1c, B2I, B2a, and B3I observations, a mixed multi-frequency UDUC BDS-2/3 PPP model was initially presented to guarantee satellite geometry and model redundancy. The positioning performance under urban environmental conditions of the BDS-2/3 was evaluated by the static and kinematic experiments. The static experiment results showed the positioning errors for BDS-2/3 could be maintained at 0.035, 0.033, and 0.218 m even when the cut-off elevation angle was 40°. Meanwhile, under simulated limited satellite conditions, the positioning performance of BDS-2/3 was better than BDS-3 in respect of positioning accuracy, convergence time and positioning continuity.

In addition, we also verified the advantages of the mixed multi-frequency model over the traditional multi-frequency model. As we can see, only the mixed multi-frequency BDS-2/3 PPP could maintain centimeter level positioning accuracy in the kinematic experiment. Compared with traditional multi-frequency BDS-2/3 PPP, the positioning errors were, respectively, reduced by 38.6, 34.5, and 46.9%, and the convergence time improved by 40.00%. The reason for the above phenomenon is that the multi-frequency model possesses the best satellite geometry and the most observation redundancy. Thus, the mixed multi-frequency BDS-2/3 PPP can achieve better positioning accuracy and convergence times under urban environmental conditions. Although the performance of the BDS-2/3 PPP under urban environmental conditions can be improved using the mixed multi-frequency model, the long convergence time is still a major challenge for the application of BDS-2/3 under urban environmental conditions.

**Author Contributions:** Conceptualization, F.Y.; methodology, F.Y. and C.Z.; software, C.Z.; validation, F.Y., C.Z., J.Z. and Z.S.; formal analysis, C.Z.; investigation, C.Z. and L.Z.; resources, L.L. All authors have read and agreed to the published version of the manuscript.

**Funding:** This research was jointly funded by the National Key Research and Development Program (No. 2021YFB3901300), the National Natural Science Foundation of China (Nos. 62003109, 61773132, 61633008, 61803115, 62003108), the 145 High-Tech Ship Innovation Project sponsored by the Chinese Ministry of Industry and Information Technology, the Heilongjiang Province Research Science Fund for Excellent Young Scholars (No. YQ2020F009), and the Fundamental Research Funds for Central Universities (Nos. 3072019CF0401, 3072020CFT0403).

**Data Availability Statement:** Not applicable.

**Acknowledgments:** The authors acknowledge IGS for providing the data.

**Conflicts of Interest:** The authors declare no conflict of interest.

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
