# Peer review of "The Initial Performance Evaluation of Mixed Multi-Frequency Undifferenced and Uncombined BDS-2/3 Precise Point Positioning under Urban Environmental Conditions"

_remotesensing, doi:10.3390/rs14215525_

Round 1

Reviewer 1 Report (Previous Reviewer 2)

The author has stated my concerns in the previous round.

Author Response

Thanks for your comments.

Reviewer 2 Report (New Reviewer)

It is of great value to study BDS PPP under urban environmental conditions. This paper analyzes the static and kinematic experiments for complex urban environments. The static experiment, using different altitude angles to analyze the positioning accuracy, confirms the visible satellite advantage of BDS-2/3; the dynamic experiment, compared with the traditional multi-frequency model, illustrates the advantages of the hybrid multi-frequency model proposed in this paper. However, there are still many inadequacies in this manuscript. I hope the following comments can help to improve the manuscript.

(1) The introduction indicates that the BDS PPP have been widely used in many fields. The status of precise satellite orbit and clock products should be added in this paper, as mentioned in the works ‘https://doi.org/10.1007/s10291-019-0834-2’ and ‘https://doi.org/10.3390/s16122192’, this is relevant to the subject of this manuscript.

(2) The meanings of IFB in line 132 and ISB in line 133 are not explained, although they are well known, it would be better to add an explanation.

(3) In Figure 3, the cut-off elevation angles should be added to each subplot.

(4) In the static experiment, the origin of the reference coordinates of these stations should be explained.

(5) In Table 4, the name of the station should be given in the title.

(6) Usually, the GNSS data-processing software package is not easy to be started from scratch. The authors should introduce the GNSS software platform involved in this manuscript.

(7) Line 304-307: ‘The observation data of the reference station and the kinematic experimental device is processed together by RTKPOST and the centimeter-level fixed-solution obtained is used as a reference to evaluate the positioning performances.’ The ‘RTKPOST’ is a component of the RTKlib software package, a brief introduction to the RTKlib software package should be added.

(8) There are some typos or grammatical errors, please check through the manuscript and polish the language. For example,

Line 298: ‘real-world data’ is not a native expression. What is the meaning of ‘real-world’? Is there another ‘fake-world’? It will be better to use ‘real data’ or ‘field data’.

Author Response

Reviewer 3 Report (New Reviewer)

Method, proposed by authors, gives good results compared with traditional multi-frequency model: the positioning accuracy of the mixed multi-frequency model improves 51.6%, 35.5%, and 39.1%, respectively in east, north and up directions. The convergence time is shortened by 40% also. Impressive numbers. What I am concerning a little bit is the conditions of experiment (Fig 8), which, from my point of view, were not so bad as it is claimed by authors. Also the number of experiments could be slightly bigger.  However that does not degrade the results achieved by authors.

So, paper could be printed.

Round 2

Reviewer 2 Report (New Reviewer)

It is a good work for this manuscript.

This manuscript is a resubmission of an earlier submission. The following is a list of the peer review reports and author responses from that submission.

Round 1

Reviewer 1 Report

Some concerns are unclear to potential readers.

1. What's the noelty of this submission? PPP model, PPP performance of static station and kinemetic test have already been discussed in previous publications. what's new in this submission?

2. what's 'specific model'? 'specific multi-freq', 'specific BDS-3'? no explaination.

3. L133. maybe you forget c, the light speed.

4. 3.2 kinematic experiment. More descriptions about the environment and the RTK reference solution are required.

L310, 3.5 hours. so, how many laps in Figure 8?

Reviewer 2 Report

1.        The introduction are much too long. Please focus on the current research state of multi-frequency undifferenced and uncombined BDS PPP under urban condition due to the decrease of the number of satellites observed. The description about BDS, performance of BDS-2, BDS-3 can be shorten with some references. 

2.        Some references about performance of GPS PPP under urban condition should be added in the introduction. 

3.        Line 83, ‘in in the vertical direction’ changed to ‘in the vertical direction’ 

4.        Formula 7 is incorrectly expressed. R is stochastic model, while P, L is observation model. 

5.        Why is the value of non-diagonal element in Formula 8 is σSISRE, which represents the correlation between P and L 

6.        In table 2, ambiguities are treated as random walk process. In many references, they are estimated as constant. Please compare the results using the above strategies. 

7.        Line 201, 30 s means sampling interval or observation length? 

8.        Line 212, what do ‘mixed multi-frequency’ and ’specific multi-frequency’ mean? It should be explained in the manuscript 

9.        Line 241, ‘convergence’ changed to ‘convergence time’ 

10.    ‘CT’ should be explained before the table 4 

11.    Line 249-250, the satellites with poor quality are excluded when cut-off elevation is 20 degree. Since the quality of the observation data is poor, why it was not exluded at 10 degree? 

12.    Line 265, positioning solutions? Positioning accuracy 

13.    Line 363, ‘where satellite and signal’ changed to ‘with satellite and signal’ 

14.    The conclusion should be shorten 

15.     There are a lot of grammatical errors. I suggest extensive editing of English language and style, or sending it for another English review by a native English speaker

Reviewer 3 Report

Dear Authors,

please, see the attachment.

Round 2

Reviewer 1 Report

Overall, the kinematic solution is still unclear to me.

Fig. 9: conparison of satellite amounts, observations and PDOPs.

Fig. 10: comparision of four PPP models.

More technical details must be provided. As I mentioned in the preview reviewer report, what's the novelty of this submission? The methodology section and Table 2 empahsize IFB and ISB. So the discussion section should support your assmuption or estimation of IFB and ISB. But, nothing concerning IFB and ISB solutions are provided. This is the weakness of your current version from my side. The link between the methodology and the discussion is very weak. 

BTW, 'specific model'. Why do you use this terminology? Is this a newly created terminology by yourself? If so, this will confuse most readers within the GNSS community. Otherwise, add the reference.

Reviewer 2 Report

1, The authors have not properly answered the 5th point raised in my first review.

The authors state in their rebuttal: Because pseudo-range and carrier phase observation data share a set of precise ephemeris and clock correction products, and σSISRE is the STDs of signal-in-space ranging errors (SISRE). According to error propagation law, the stochastic model of single satellite at single frequency based on Equation (1) can be showed as follows …”

Some remarks: The pseudo-range and carrier phase observations share several common errors, such as orbit and clock error, receiver clock error, atmospheric signal error, relativistic effects, etc. But they are the systematic errors, not random noises. Stochastic model is based on the random noises, and the error propagation law is also true for random noises. The common systematic errors have nothing to do with the random noises of pseudo-range and carrier phase observations. So, I do not agree that the correlation between P and L is the expression of σSISRE. Suggest that the correlation between P and L is changed to zero, and the variance of P and L remove the σSISRE. 

2, Line 21, “five frequency BDS-2/3 stations” bad English

Reviewer 3 Report

Dear Authors,

Thank you very much for addressing all my comments. The paper has been improved and I think that now it can be accepted for publication.

Author Response

Thank you very much for all of your comments.